# Radiomic Analysis of MRI Images is Instrumental to the Stratification of Ovarian Cysts

**DOI:** 10.3390/jpm10030127

**Published:** 2020-09-14

**Authors:** Roxana-Adelina Lupean, Paul-Andrei Ștefan, Diana Sorina Feier, Csaba Csutak, Balaji Ganeshan, Andrei Lebovici, Bianca Petresc, Carmen Mihaela Mihu

**Affiliations:** 1Histology, Morphological Sciences Department, “Iuliu Hațieganu” University of Medicine and Pharmacy, Louis Pasteur Street, number 4, Cluj-Napoca, 400349 Cluj, Romania; roxanalupean92@gmail.com (R.-A.L.); carmenmihu@umfcluj.ro (C.M.M.); 2Obstetrics and Gynecology Clinic “Dominic Stanca”, County Emergency Hospital, 21 Decembrie 1989 Boulevard, number 55, Cluj-Napoca, 400094 Cluj, Romania; 3Anatomy and Embryology, Morphological Sciences Department, “Iuliu Haţieganu” University of Medicine and Pharmacy, Victor Babeș Street, number 8, Cluj-Napoca, 400012 Cluj, Romania; 4Radiology and Imaging Department, County Emergency Hospital, Cluj-Napoca, Clinicilor Street, number 5, Cluj-Napoca, 400006 Cluj, Romania; csutakcsaba@yahoo.com (C.C.); andrei1079@yahoo.com (A.L.); bianca.petresc@gmail.com (B.P.); 5Radiology, Surgical Specialties Department, “Iuliu Haţieganu” University of Medicine and Pharmacy, Clinicilor Street, number 3-5, Cluj-Napoca, 400006 Cluj, Romania; 6Institute of Nuclear Medicine, University College London Hospitals NHS Trust, 235 Euston Road, London NW1 2BU, UK; b.ganeshan@ucl.ac.uk

**Keywords:** computer-aided diagnosis, disease modeling, magnetic resonance imaging (MRI), ovarian cyst, patient stratification, personalized medicine, prediction, texture analysis

## Abstract

The imaging diagnosis of malignant ovarian cysts relies on their morphological features, which are not always specific to malignancy. The histological analysis of these cysts shows specific fluid characteristics, which cannot be assessed by conventional imaging techniques. This study investigates whether the texture-based radiomics analysis (TA) of magnetic resonance (MRI) images of the fluid content within ovarian cysts can function as a noninvasive tool in differentiating between benign and malignant lesions. Twenty-eight patients with benign (*n* = 15) and malignant (*n* = 13) ovarian cysts who underwent MRI examinations were retrospectively included. TA of the fluid component was undertaken on an axial T2-weighted sequence. A comparison of resulted parameters between benign and malignant groups was undertaken using univariate, multivariate, multiple regression, and receiver operating characteristics analyses, with the calculation of the area under the curve (AUC). The standard deviation of pixel intensity was identified as an independent predictor of malignant cysts (AUC = 0.738; sensitivity, 61.54%; specificity, 86.67%). The prediction model was able to identify malignant lesions with 84.62% sensitivity and 80% specificity (AUC = 0.841). TA of the fluid contained within the ovarian cysts can differentiate between malignant and benign lesions and potentially act as a noninvasive tool augmenting the imaging diagnosis of ovarian cystic lesions.

## 1. Introduction

Due to the complexity of their embryological and histological development, the ovaries represent a source of a wide variety of lesions [1]. Ovarian cysts represent a common gynecological encounter as a consequence of physiological or pathological processes [2].

The imaging characterization of ovarian cysts is of great importance to plan adequate therapeutic procedures and strongly influence the patient’s management [3]. Although transvaginal ultrasonography (TVUS) is the first-line imaging modality used to characterize adnexal masses [4], a proportion of them remains indeterminate, mostly regarding their benign or malignant character [5]. Magnetic resonance imaging (MRI) is an important diagnostic tool that can offer accurate information about the features and the local invasion of a pelvic mass [6]. Commonly used MRI sequences in this scenario include nonenhanced T1- and T2-weighted sequences [6], but also advanced techniques such as diffusion-weighted (DWI) [7] and dynamic contrast-enhanced imaging [8].

The pathological analysis of ovarian cysts shows that the fluid composition can be specific for certain adnexal lesions [9] in terms of physical properties (such as density, color, homogeneity), biochemical compounds, and cellularity [10]. On T2-weighted images (T2WIs), the fluid appearance may vary from high signal (predominantly pure liquid) to an intermediate or low signal (in case of an old hemorrhage or contamination) [3]. However, the fluid appearance on imaging rarely plays an important role in differentiating among various types of lesions [11], and for this reason, physicians often pay more attention to the morphological signs linked to malignancy (such as cystic septations and internal vegetations) [3,12]. However, these morphological features can sometimes be misleading; different histopathological entities may show similar radiologic characteristics [3].

It is hypothesized that the specific fluid properties may be reflected in MRI, altering the pixel intensity and distribution, but their influence on signal intensity is too subtle to be visually appreciated. Texture analysis (TA) is a technique that provides quantitative information about image pixel intensity and distribution (subtle heterogeneity not perceptible to the naked eye), and in recent years, it has been increasingly used in radiology research as part of radiomics, machine-learning, and artificial intelligence techniques within computer-aided diagnosis [13]. By defined parameters, TA can offer information about tissue characteristics, and its utility has been proven, especially in the diagnosis, disease-severity, treatment-response, prediction, and prognosis of several tumor sites [14].

In this pilot study, we quantified the texture features of the fluid contained in several types of ovarian cysts by applying texture-based radiomics analysis of T2WI. The purpose is to investigate whether the subtle differences in fluid composition are reflected in the texture parameters extracted from MRI images and if TA-based features can be used to distinguish malignant from benign ovarian cysts.

## 2. Materials and Methods

### 2.1. Patients

This Health Insurance Portability and Accountability Act (HIPAA)-compliant, single-institution, retrospective pilot study has been approved by the institutional review board and ethics committee of the “Iuliu Hațieganu” University of Medicine and Pharmacy Cluj-Napoca (Registration no. 50, 11 March 2019), and written informed consent was waived by all subjects, owing to the retrospective nature of the study using anonymized patient images and data. From May 2017 to March 2020, a keyword search using the terms “ovary + cyst/s”, “ovarian + cyst/s”, and “adnexal” was conducted in our radiology information system (RIS) to identify all MRI pelvic examinations referring to ovarian cystic lesions. The original search yielded 221 results. Each report was then analyzed by a researcher, and all cases in which the keyword in the report did not refer to an ovarian cystic lesion were excluded (*n* = 14). The medical records of the remaining 207 patients were retrieved from the archive of our healthcare unit and investigated for disease-related data. The other exclusion criteria were artifacts affecting the T2-weighted (T2W) sequences (*n* = 8), cysts identified by other imaging protocols than the dedicated pelvic protocol (*n* = 20), massive fluid contamination, as seen on T2WI or described in the pathological analysis (*n* = 13), a minimum diameter of the fluid component of at least 15 mm (*n* = 41), no gynecological follow-up (*n* = 36), and lack of a final clinical or pathological diagnosis (*n* = 61). The final study population comprised of 28 patients.

### 2.2. Reference Standard

For pathological analysis, microscopic sections were obtained from the largest tumor dimension. For each lesion, one-to-three cell blocks were prepared, stained with hematoxylin and eosin, and analyzed microscopically. Twenty-two cysts also beneficiated from a gross description of their content, but cytological analysis of the fluid was not performed on any lesion. All ovarian serous cystadenomas, clear-cell carcinomas, and high-grade serous carcinomas were surgically removed and were pathologically analyzed. All functional cysts (FCs) represented incidental findings in the MRI examinations. Four FCs that belonged to ovaries were surgically removed, and they underwent pathological analysis along with the underlying disease. Six FCs were followed by gynecologists along with their underlying disease (uterine fibroma, *n* = 2; cervical malignancy, *n* = 2; endometrial polyps, *n* = 1), and their remission in time was noted (mean time from MRI to remission: 156.3 days, range: 127–189 days; Table 1).

### 2.3. Imaging Protocol

All scans were performed on the same scanner (1.5T, SIGNA™ Explorer, General Electrics, Waukesha, WI, USA) using a multicoil phased array. No routine antiperistaltic drugs were used, but instead, a narrow band around the abdomen was applied to diminish the intestinal peristalsis. The acquisition protocol consisted of axial T1, sagittal T2 Periodically Rotated Overlapping Parallel Lines with Enhanced Reconstruction (PROPELLER), oblique axial T2 Fast Relaxation Fast Spin Echo (FR-FSE) High Resolution (HR), oblique coronal T2 FR-FSE HR, axial diffusion-weighted sequences (DWI), and contrast-enhanced T1 fat-suppressed. The imaging protocol varied because the examinations were selected from a range of about 3 years, but each pelvic MRI examination consistently comprised an oblique axial T2 sequence, which was the only sequence used in this study (repetition time/echo time: 4556/109 ms; refocus angle: 160; section thickness: 4 mm; slice spacing: 0.4 mm; bandwidth: 83.333 Hz/pixel; field of view: 240 mm; matrix: 320 × 320; slices: 37; acquisition time: 323 s) for texture analysis.

### 2.4. Texture Analysis

On a dedicated workstation (General Electric, Advantage workstation, 4.7 edition), all examinations were reviewed by one radiologist and one gynecologist (R.-A.L. and A.L.) who were aware of the patients’ final pathological and clinical outcomes. When multiple ovarian cysts were observed within the same examination, the images were cross-referenced with the pathological and TVUS results and other medical data to ensure the selection of lesions that were previously documented. From the oblique axial T2W sequence of each examination, only one cyst was selected and marked. All examinations were anonymized, and the oblique axial T2W sequences were retrieved in DICOM format (Digital Imaging and Communications in Medicine).

The classic approach of radiomics consists of four steps: image acquisition, identification of regions of interest and image segmentation, feature extraction and analysis, and prediction [15]. The image acquisition has been described above. For image segmentation, a second researcher (C.C., with 15 years of experience in pelvic MRI, blinded to the final diagnosis) reviewed the retrieved sequence and, from every examination, selected a single slice with the largest cross-section area of the previously selected lesion (considered to be representative for the cystic component). Selected images were exported in DICOM format and imported into commercially available research software, TexRAD (Feedback Medical Ltd., Cambridge, UK). The same researcher free-handedly drew a region of interest (ROI) that incorporated the fluid content within the platform for texture analysis, avoiding lesion walls, intracystic proliferations, or other solid components.

The feature extraction step was performed within the TexRAD software. Texture analysis assessed the heterogeneity within each ROI in the T2WI using a filtration-histogram-based textural analysis technique. The filtration step, using a band-pass Laplacian of Gaussian (LoG) filter (similar to a nonorthogonal wavelet), comprised of extracting and enhancing image features of different sizes and intensity variation corresponding to the spatial scale of the filter (SSF in radius). The feature scales used ranged from SSF = 2–6 mm, where the fine texture scale corresponds to SSF = 2 mm, the medium texture scale corresponds to SSF = 4 mm, and the coarse texture scale corresponds to SSF = 6 mm. Following the filtration step, quantification of texture was undertaken using statistical- and histogram-based parameters at each derived (filter-scale, SSF value) image, as well as on the conventional image (without filtration, SSF = 0). Statistical and histogram parameters comprised of mean, standard deviation of pixel intensity (SD), mean of positive pixels (MPP), skewness, and kurtosis. Figure 1 provides a visual illustration of the filtration histogram based on a T2WI of an ovarian cyst (Figure 1).

### 2.5. Statistical Analysis

The final step of the radiomics approach consists of analysis and prediction. Firstly, the univariate, multivariate, and the receiver operating characteristic (ROC) analyses were used to assess the texture features’ ability to distinguish between different groups and subgroups of patients with ovarian cysts. The nonparametric Kruskal–Wallis test assessed the ability of MRI texture parameters to differentiate between the patient subgroups (serous carcinoma, clear-cell carcinoma, serous cystadenoma, and functional cyst). For differentiating between benign and malignant groups, the absolute value of each texture parameter computed with and without filtration was compared using a nonparametric test of the null hypothesis (Mann–Whitney U-test). A *p*-value of less than 0.05 was considered statistically significant. For the most significant texture markers differentiating between malignant and benign lesions, diagnostic criteria were established by ROC analysis, where the area under the ROC curve (AUC), sensitivity, and specificity were calculated and plotted with 95% confidence intervals (CIs). A comparison of ROC curves was conducted using the DeLong et al. method. Secondly, multiple regression analysis was performed using an “enter” input model to identify which of the texture parameters that showed statistically significant results at the univariate analysis are also independent predictors of malignant lesions. The coefficient of determination (R-squared) was computed, and the diagnostic value of the prediction model was evaluated using ROC analysis. Statistical analysis was performed using a commercially available dedicated software, MedCalc version 14.8.1 (MedCalc Software, Mariakerke, Belgium).

## 3. Results

Of the 221 patients with adnexal lesions referred to our department during the study period, the final study population (after applying the inclusion and exclusion criteria) comprised of 28 patients (average age ± standard deviation: 39.28 ± 12.78 years; age range: 24–78 years). Patients were divided according to the final diagnosis of their disease into benign (*n* = 15) and malignant cysts groups (*n* = 13). The benign group comprised of five cystadenomas and ten FCs, while the malignant group incorporated five serous carcinomas and eight clear cell carcinomas.

From every image, six parameters computed for four SSF values were extracted, totaling 24 texture parameters. The multivariate analysis showed statistically significant results for the mean extracted from every filter scale, entropy, and SD extracted from raw images, kurtosis from two filter scales, and MPP and skewness from three filter scales (Table 2, Figure 2).

The univariate analysis showed statistically significant differences between benign and malignant cysts for SD computed from the medium texture level (SSF = 4; *p* = 0.033) and for skewness (*p* = 0.017) and kurtosis (*p* = 0.002) computed from raw images (Table 3). The median values of these parameters for each group are shown in Table 4.

The multiple regression analysis that integrated all parameters showing statistically significant results in the univariate analysis showed that SD (SSF = 4; *p* = 0.03) was the only independent predictor for malignant lesions (Table 5). The variance inflation factor (VIF) yielded low values for all texture parameters, which rejected the hypothesis of the multicollinearity of the independent variables.

The ROC analysis results indicated that kurtosis computed from raw images had the highest AUC of all individual parameters (0.836). AUC showed by the prediction model (0.841) was not statistically different from the ones exhibited by the three parameters comprised in the model (*p* = 0.29, 0.3, and 0.9). The ROC curve comparison showed statistically significant results between skewness and kurtosis, but not between SD and the premiums. The prediction model’s sensitivity was equal, but the specificity was lower than the ones showed by skewness and kurtosis (Table 6, Figure 3).

## 4. Discussion

### 4.1. Study Outcomes

It was previously documented that the ovarian cysts’ fluid content shows particularities for certain histopathological groups in terms of liquid [9,16] and cellular density [17], cellular population [10,18], and biomarkers [19,20]. Three of the four included pathologically proven FCs had a yellow color, an appearance that has been previously attributed to the accumulation of bilirubin [17].

Our univariate analysis showed that SD computed from the medium texture level (SSF = 4; *p* = 0.033) and the raw extraction of skewness (*p* = 0.017) and kurtosis (*p* = 0.002) have the ability to successfully differentiate between benign and malignant ovarian cysts. The SD parameter is a measure of the variation from the average value of pixel intensity. A high SD indicates that the data points are extended over a wide value range [13]. Serous carcinomas present greater cellularity than seen in aspirates of serous cystadenomas [17]. This high cellularity, together with the fluctuating appearance of clear-cell carcinomas (Table 1), could cause variations in signal intensities in T2WI, generating a larger spread of intensity “values” that are reflected in the higher values of the SD parameter for the malignant group. The skewness parameter measures histogram asymmetry [13], which follows the degree of imbalance in the distribution of gray-level intensities around the mean. The negative values of this parameter indicate that the histogram distribution is dense toward the high gray-level intensities (bright images) [21]. We observed a lower value of skewness in the benign group (Table 4), which suggests that these lesions exhibit a higher signal intensity in the T2WI, probably due to their content of less contaminated fluid.

Our results also showed that the TA parameter kurtosis, computed from unfiltered MRI images, had a high diagnostic ability to identify malignant lesions (AUC = 0.836; sensitivity: 84.62%; specificity: 93.33%). This parameter is a measure of the peakedness of the histogram. If the distribution of the gray-levels around the mean pixel value is flat (e.g., low values of kurtosis/wider histogram), it implies that the processed image has more information [20]. We observed lower values of this parameter for the malignant group, probably because the fluid content is much more heterogeneous than in the benign group. Another reason for the successful differentiation between the two groups could be represented by the similar fluid appearance of the two entities included in the benign group. The granulosa cells contained in FCs resemble macrophages [16], the latter being found in abundance in serous benign tumors [22]. Additionally, for a large proportion of both entities, a yellow fluid was described in the pathological analysis. However, it is not clear which of the cystic features were reflected in the values of the TA parameters, since both physical and histological characteristics could influence fluid appearance in T2WIs.

### 4.2. Limitations of the Current Imaging and Cytological Diagnosis of Ovarian Malignancies

Many studies in the literature focus on the distinction between benign and malignant ovarian lesions based on imaging features. MRI has proved useful in determining the site of origin of a pelvic tumor, characterizing adnexal masses, and detecting the local invasion [3]. However, a meta-analysis [23] that investigated the value of DWI in the differential diagnosis of adnexal masses concluded that quantitative DWI is not a reliable method for differentiation between benign and malignant ovarian masses. On the other hand, based on MRI morphological features, benign and malignant ovarian tumors were discriminated with 88–93% accuracy in one study [24] and with 100% sensitivity and 95% specificity in another [25]. However, the latter research [25] included multiple morphological types of ovarian lesions (solid, cystic, and mixed-types), the benign group also comprised of nontumoral entities (tubo-ovarian abscess and hematosalpinx), and the differentiation was based on both the lesions aspect as well as ancillary findings (such as lymph node involvement). The same research [25] showed that when evaluating only cystic lesions, MRI failed to distinguish between benign and malignant cysts in 25% of the cases, even after considering the ancillary features. A possible explanation may be due to imaging features such as septa, multilocularity, and wall thickening, which have low reliability for the diagnostic of malignancy because these features are often seen in benign tumors (such as cystadenomas, fibromas, and endometriomas) [26]. These observations create the need to improve the MRI diagnosis of ovarian cysts. We were unable to find any study in the literature comparing the imaging characteristics of ovarian cysts’ fluid. This may be due to possible contamination (especially with blood) that may accompany the evolution of these lesions, leading to a fluctuation in measurements [27], therefore providing inconsistent data.

Laparoscopy and laparotomy remain the most accurate methods for the diagnosis of malignant ovarian lesions [28]. However, these techniques also predispose to bowel perforations and laceration and damage of other organs and vessels that could lead to hemorrhage [29]. Even the ultrasound-guided fine-needle aspiration (FNA) procedure often causes pain and is linked to hematuria due to the transvesical approach of the needle [28,30]. Additionally, the latter technique has never been widely accepted by physicians, mostly due to the fear of spreading malignant cells in case of unrecognized malignancy and also due to the low sensitivity provided by cytological analysis of the fluid samples [31]. Moreover, FNA management is beneficial only for simple unilocular cysts in premenopausal women without ultrasound evidence for malignancy and with clear aspirate [27,28]. There are also a series of limitations regarding the cytological analysis of the fluid sampled by FNA. De Crespigny [30] observed that 74% of the fluid samples had acellular sediment or contained blood cells or histiocytes, and no specific prediction of cyst type could be made. This observation is supported by another study conducted by Diernaes [31], which showed that cytology was unable to provide a diagnosis in 5 out of 7 malignancies in their series. Moreover, research conducted by Mulvany [16,17] showed that almost 56% (131/235) of the aspirates were devoid of diagnostic cells. Overall, the possibility of cystic rupture and the frequency of inadequate cell harvest impose discretion on the use of this method [17]. Apart from FNA, the diagnostic power of cytology varies along with other sampling methods. Several studies investigated the ability of cytology to detect malignant cells from fluids sampled using laparoscopy or laparotomy. The sensitivity ranged from 26–50% and the specificity from 76–100%, but the study population and fluid analysis procedure were unalike [32,33].

### 4.3. Future Perspectives

There is an obvious need for a more accurate diagnosis of cystic ovarian lesions, preferably noninvasive. If quantitative measurements extracted from MRI images of ovarian cysts could be further linked to the cellularity comprised in these lesions, it would bring enormous benefit to current medical practice and relieve patients of other invasive interventions. As the imaging morphological features advocating the malignant nature of an ovarian cyst appear in the later stages of the disease and because early ovarian malignancies lack diagnostics biomarkers, there is an immense need to identify novel diagnostic approaches [34]. Such markers could be provided by TA analysis and other radiomics models. Previous radiomics applications in oncology have demonstrated that TA parameters are linked to tumoral phenotypic patterns [35], chemoradiotherapy response [36], distant metastasis and survival [37], and many others [38]. Thus, it is possible that these parameters could also reflect the genomic profile of ovarian malignancies. This approach may be useful, especially in tumors with high heterogeneity where biopsy is unable to provide sufficient tissue for whole-tumor genomic state analysis or when the biopsy is contraindicated [15]. If further validated, the radiomics application could become a noninvasive method of assessing the tumoral genome, thus innovating the concept of personalized oncology [38]. Moreover, malignant ovarian cysts contain many protein biomarkers that may be used for cancer detection [39], such as protein C inhibitor, apolipoprotein C-I and C-III, serum amyloid 4, and transthyretin [40]. Thus, it is possible that the combination of radiomics and cytological biomarkers could provide adequate information that would significantly improve the stratification, prediction, and treatment of oncological patients.

### 4.4. Socioeconomical Aspects

The socioeconomical aspect of adnexal cysts should not be neglected. The diagnosis of an ovarian mass causes anxiety in patients, which often pressures physicians to remove it out of fear that the patients have cancer [39]. This unnecessary surgery represents a significant cost to society and also to patients because it increases the risk of ectopic pregnancies and may interfere with fertility [41]. On the other hand, MRI examinations also do not come at a low cost. However, our results suggest that the use of TA on a routinely acquired T2W sequence is less time-consuming and comes at a lower cost than the contrast-enhanced and diffusion-weighted MRI pelvic examinations. If furtherly validated, this approach can function as a complementary tool to pelvic MRI examinations on patients with indeterminate ultrasound lesions, but only if TA parameters can be directly linked to the tumoral cellularity comprised in the cystic fluid.

### 4.5. Study Limitations

Our study had several limitations. Due to its retrospective nature, there may have been selection bias. We preferably selected only lesions that were not complicated by bleeding to allow the evaluation of the true texture properties of the fluid. Thus, it is possible that major blood contamination would affect this analysis since blood degradation products can decrease the signal in T2WI. Additionally, due to the strict inclusion and exclusion criteria of this pilot study and the overall limited number of cases that were referred to our department in the study period, we included very few distinct histopathological entities in each group. Moreover, the overall number of patients was relatively small. The ROI segmentation employed in this pilot study comprised a single largest cross-section-based delineation instead of a multislice or three-dimensional volume analysis. However, previous studies using the filtration-histogram technique have demonstrated the comparison of single-slice vs. multislice/volume analysis on computer tomography (CT) in primary colorectal cancer for prognostication [42], as well as on MRI in gliomas for IDH versus wild-type differentiation [43]. Interestingly the analysis demonstrated that single-slice analysis was significant in predicting prognosis in colorectal cancer on CT [42] and IDH vs. wild-type differentiation in gliomas on MRI [43], and it was comparable to multislice/volume analysis. It is not, therefore, clear if there is any “significant” added-value of undertaking multislice/volumetric analysis, which not only entails increased analysis time (barrier to adoption in a busy clinic) but also increased operator variability associated with multislice/volume analysis. Another limitation can potentially come from the fact that no intra- or interobserver agreement was assessed. However, previous studies on CT and MRI have demonstrated good reproducibility for filtration-histogram-based TA using multicenter clinical validation [37,38], robustness to variation in image acquisition parameters [44,45,46,47], and good inter- and intraoperator repeatability (good intraclass correlation from test–retest technique) [48,49]. The lack of cytological analysis of the cystic fluid can be considered a major limitation. However, previous research showed that cytology has low sensitivity in detecting malignant cells, regardless of the sampling method [17,31]. Probably a more extensive analysis that could also reflect the biochemical features of the fluid would be equally useful, but unfortunately, it is not routinely used in our center. The fact that two researchers (R.-A.L. and A.L.) were aware of the final diagnosis can also be viewed as a limitation. However, this approach was necessary because at the time of the MRI examinations, several patients had multiple lesions, and we desired to include only the ones that were pathologically or at least clinically documented. After this stage, these investigators (R.-A.L. and A.L.) did not intervene in any way in interpreting the images, reporting the results, or conducting the statistical analysis. The TA software used in our study allowed the computation of only histogram-based texture features. We acknowledge that the use of second- and higher-order statistics may provide a more extensive description of these lesions’ fluid content. As the latter texture features could not be generated, our study invites future research to validate this method by another TA program that allows the computation of a larger number of parameters.

## 5. Conclusions

The results of our pilot study demonstrate that our texture-based radiomics methodology could be beneficial to patients with ultrasonic-indeterminate ovarian cysts, not complicated by bleeding, but only if a direct relationship between tumoral cells and the TA parameters can be further demonstrated. Although texture parameters yield the ability to successfully discriminate between benign and malignant ovarian cystic tumors, it is unclear whether these parameters reflect the biochemical features of the lesion’s content or their appurtenance of a certain histopathological group.

## Figures and Tables

**Figure 1 jpm-10-00127-f001:**
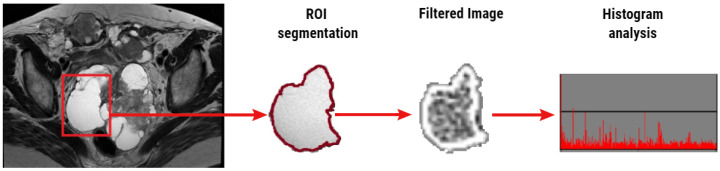
Illustration (workflow) of the texture analysis of a T2-weighted image (T2WI) based on the filtration histogram technique. The conventional T2WI (left) of a 61-year-old patient with pathologically proven clear cell carcinoma is segmented using a freehand region of interest (ROI; red square). The distribution of fluid texture features within the conventional and filtered image was assessed using texture parameters derived from statistical- and histogram-based analysis.

**Figure 2 jpm-10-00127-f002:**
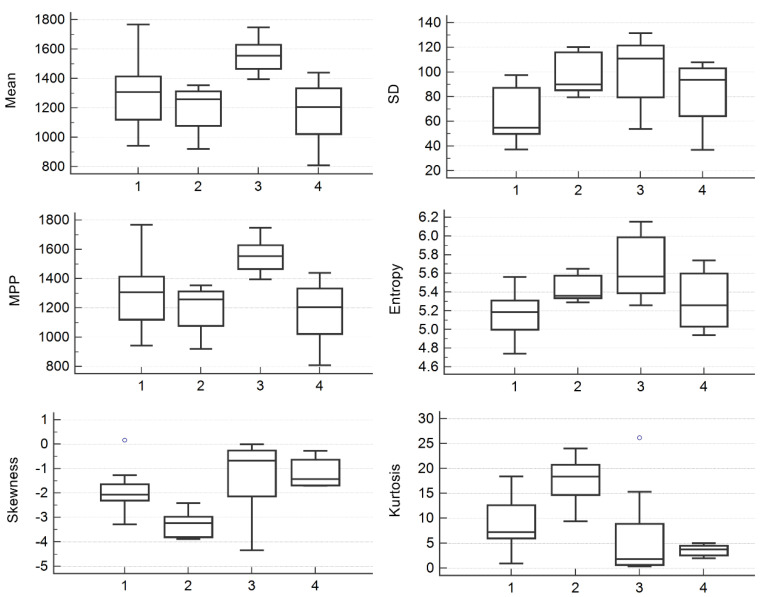
Box and whisker plots showing differences between median values of mean, SD, MPP, entropy, skewness, and kurtosis computed from raw images for the differentiation of functional cysts (1), serous cystadenomas (2), clear-cell carcinomas (3), and serous carcinomas (4). SD, standard deviation of pixel intensity; MPP, mean of positive pixels.

**Figure 3 jpm-10-00127-f003:**
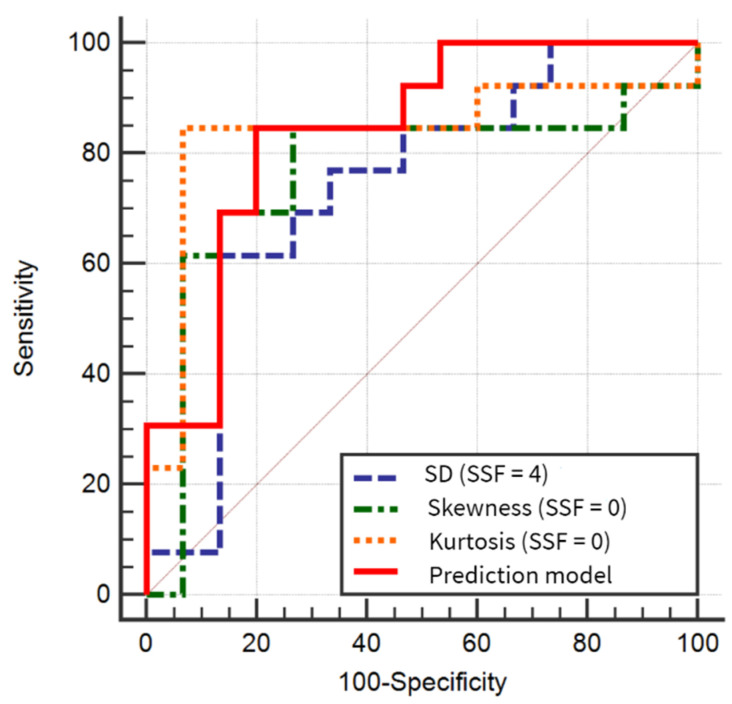
Comparison of areas under the curves for the differentiation of malignant from benign cysts, based on the three texture parameters that showed statistically significant results in the univariate analysis and the prediction model. SSF, the spatial scale of the band-pass filter; SD, standard deviation of pixel intensity.

**Table 1 jpm-10-00127-t001:** Pathological analysis.

Type of Lesion	Number of Lesions	Pathological Confirmation		Pathological Findings
		Lesions/Total	Time (Days)	
High-grade serous carcinoma	5	5/5	64.4 ± 31.8	serous fluid, 20% *; gelatinous content, 20%; brownish fluid, 40%; turbid liquid, 40%.
Clear cell carcinoma	8	8/8	96.2 ± 21.5	clear liquid, 75%; slightly hemorrhagic fluid, 25%.
Serous cystadenoma	5	5/5	62.1 ± 34.3	clear liquid, 60%; yellow turbid, 40%.
Functional cysts	10	4/10	48.7 ± 18.5	yellow fluid, 75%; serous fluid 25%

* percentage of lesions with the same findings from each subgroup.

**Table 2 jpm-10-00127-t002:** Multivariate analysis results.

SSF	Texture Parameter					
	Mean	SD	MPP	Entropy	Skewness	Kurtosis
0	**0.01**	**0.027**	**0.01**	**0.02**	**0.027**	**0.009**
2	**0.025**	0.082	0.053	0.683	0.375	0.017
4	**0.016**	0.119	**0.018**	0.455	**0.034**	0.088
6	**0.019**	0.507	**0.021**	0.433	**0.028**	0.341

Statistically significant results of the Kruskal–Wallis test are highlighted in bold. SSF, the spatial scale of the band-pass filter; SD, standard deviation of pixel intensity; MPP, mean of positive pixels.

**Table 3 jpm-10-00127-t003:** Univariate analysis results for comparing benign and malignant groups.

SSF	Texture Parameter					
	Mean	SD	MPP	Entropy	Skewness	Kurtosis
0	0.13	0.088	0.13	0.072	**0.017**	**0.002**
2	0.235	0.201	0.201	0.683	0.892	0.751
4	0.217	**0.033**	0.201	0.525	0.786	0.786
6	0.387	0.294	0.363	0.65	0.44	0.44

Statistically significant results of the Mann-Whitney U-test are highlighted in bold.

**Table 4 jpm-10-00127-t004:** The median values of the parameters that showed statistically significant results in the univariate analysis. In brackets, values corresponding to the interquartile range.

Texture Parameter	Benign Group	Malignant Group
SD (SSF = 4)	446.05 (363.97–501.23)	611.05 (476.09–664.54)
Skewness (SSF = 0)	−2.31 (−3.21 to −1.76)	−0.92 (−1.68 to −0.37)
Kurtosis (SSF = 0)	9.41 (6.46–18.26)	2.41 (1.43–4.46)

**Table 5 jpm-10-00127-t005:** Multivariate analysis of parameters independently associated with the presence of malignant cysts.

Parameter	Coefficient	Standard Error	*p*-Value	VIF
**SD**	0.001	<0.001	**0.03**	1.03
**Skewness**	0.202	0.194	0.3	8.89
**Kurtosis**	0.004	0.032	0.9	8.9
**Sign.lvl.**	0.016			
**R^2^**	0.342			
**R^2^ adjusted**	0.26			
**M.C. Coeff**	0.585			

Bold values are statistically significant. VIF, variance inflation factor; R^2^, coefficient of determination; R^2^ adjusted, coefficient of determination adjusted for the number of independent variables in the regression model; Sign.lvl., significance level of the multivariate analysis; M.C. Coef., multiple correlation coefficient.

**Table 6 jpm-10-00127-t006:** Receiver operating analysis results for the differentiation of malignant from benign lesions. Each *p*-value column represents the comparison between all parameters and the reference one (REF).

Texture Parameter	AUC	Sign. Level	J	Cut-off Value	Sensitivity(%)	Specificity(%)	*p*-Value	*p*-Value	*p*-Value	*p*-Value
SD (SSF = 4)	0.738 (0.539–0.885)	0.0167	0.48	>528.57	61.54 (31.6–86.1)	86.67 (59.5–98.3)	REF	0.87	0.5	0.29
Skewness (SSF = 0)	0.746 (0.567–0.903)	0.013	0.57	>(−2.07)	84.62 (54.6–98.1)	73.33 (44.9–92.2)	0.87	REF	**0.03**	**0.3**
Kurtosis (SSF = 0)	0.836 (0.648–0.948)	0.0003	0.77	≤5	84.62 (54.6–98.1)	93.33 (68.1–99.8)	0.5	**0.03**	REF	0.9
Prediction model	0.841 (0.654–0.951)	<0.0001	0.64	>0.4186	84.62 (54.6–98.1)	80 (51.9–95.7)	0.29	0.3	0.9	REF

AUC, area under the curve; Sign. level, significance level; J, Youden index. Each *p*-value column represents the comparison between all parameters and the reference one (REF). Bold values are statistically significant. Values between the brackets correspond to a 95% confidence interval.

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
