# Peer review of "Radiomic Analysis of MRI Images is Instrumental to the Stratification of Ovarian Cysts"

_jpm, 2020, doi:10.3390/jpm10030127_

Round 1
Reviewer 1 Report
General comments
The manuscript is well prepared. The study design is very thorough. The authors are well aware of the limitation of the current study. The results are clear, indicating promising results of the proposed approach.
Major comments
- The authors used TexRAD for the radiomics analysis. But, you calculated only the histogram based features (the first order statistics). Please comment the reasons for the feature selection.
- The authors used a “prediction model”. From the text, I assume that the model was built by applying a multiple regression analysis. I think you need to add more details about the prediction model. Is it in TexRAD? Was the model training done? If so, how was it performed? Did you use all 28 data for training? The ROC results are for the training?
Reviewer 2 Report
The manuscript presents valubale and clinically relevant data. However, revisions proposed below would significantly increase the overall quality of the publication.
- The title should be reconsider in order to provide clear message resulted from the study such as "Radiomic Analysis of MRI images is instrumental to stratify Ovarian cysts".
- Keywords should be extended to support reader interst in this publication such as "patient stratification", "prediction", "disease modelling", "personalised medicine" etc.
- Consequently these items should be detailed in the paper. Doing this the authors are kindly asked to refer to the relevant papers such as: Identification of clinical trait-related lncRNA and mRNA biomarkers with weighted gene co-expression network analysis as useful tool for personalized medicine in ovarian cancer. doi: 10.1007/s13167-019-00175-0;
- Further, the concept of multi-omic biomarkers (e.g. radiomics combined with liquid biopsy) should be mentioned for better patient stratification as well as cost-effective cancer prediction and personalised treatment algorithms: The crucial role of multiomic approach in cancer research and clinically relevant outcomes. doi: 10.1007/s13167-018-0128-8; Preventive, predictive, and personalized medicine for effective and affordable cancer care. 2018 doi: 10.1007/s13167-018-0130-1.
- Discussion should be better structured providing subtitles such as "Study outcomes", Socio-economical aspects; Study limitations
Round 2
Reviewer 2 Report
Congratulations to the well written paper.